# 1 Degradation of anhydro-saccharides and the driving factors

## in real atmospheric conditions: A cross-city study in China

- Biao Zhou <sup>a</sup>, Kun Zhang <sup>a</sup>, Qiongqiong Wang <sup>b</sup>, Jiqi Zhu <sup>a</sup>, Li Li <sup>a, \*</sup> and Jian Zhen Yu <sup>c,d, \*</sup>
- a School of Environmental and Chemical Engineering, Shanghai University, Shanghai, 200444,
- China
- b Department of Atmospheric Science, School of Environmental Studies, China University of
- Geosciences, Wuhan, China
- 8 ° Department of Chemistry, The Hong Kong University of Science & Technology, Hong Kong,
- China

12

14 15

- 10 d Division of Environment & Sustainability, The Hong Kong University of Science & Technology,
- 11 Hong Kong, China

13 Correspondence to: Li Li (Lily@shu.edu.cn) and Jian Zhen Yu (jian.yu@ust.hk)

## Abstract

Anhydro-saccharides (levoglucosan, mannosan, and galactosan), as important components of organic aerosol, have been widely used as molecular markers for biomass burning. Previous studies have shown that levoglucosan degrades in the atmosphere, but most of the results are derived from laboratory experiments, little is known about the decay rates and their driving factors in the real complex ambient environment. In this study, a Thermal Desorption Aerosol Gas Chromatography-Mass Spectrometry (TAG-GC/MS) was utilized to collect PM<sub>2.5</sub>-bound saccharides in three typical cities across the major city clusters in eastern China (Zibo, North China Plain; Changzhou, Yangtze River Delta; and Hong Kong, Pearl River Delta region) during the autumn and winter seasons, with bihourly time resolution. With the relative rate constant method, we found the daytime (8:00–16:00 LST) decay rate of levoglucosan was fastest in Changzhou, reaching  $0.13 \pm 0.05 \, h^{-1}$  (with a range of  $0.01\sim0.55 \, h^{-1}$ ), and the maximum decay rates of mannosan  $(0.14 \pm 0.05 \, h^{-1}$ , range:  $0.04\sim0.29 \, h^{-1}$ ) and galactosan  $(0.15 \pm 0.06 \, h^{-1}$ , range:  $0.04\sim0.33 \, h^{-1}$ ) were observed in Hong Kong. Results from the generalized additive model indicate that the daytime decay rate of anhydro-saccharides is primarily influenced by aerosol liquid water content, relative humidity, and atmospheric oxidation

1

- capacity, while temperature and solar surface radiation also contribute to an increase in the decay
- rates. This study provides valuable field data on the degradation rates of saccharides in real ambient
- environments and demonstrates that their degradation results are derived by the combined effects
- of multiple oxidation pathways.
- **Keywords:** Anhydro-saccharides; Degradation rates; Real atmosphere; TAG-GC/MS
- Highlights
- Anhydro-saccharides (levoglucosan, mannosan, and galactosan) exhibit decreasing trends
- during the daytime in ambient environment.
- The decay rate of levoglucosan is fastest in Changzhou, while the decay rates of mannosan and
- galactosan are fastest in Hong Kong.
- The daytime decay of anhydro-saccharides is primarily influenced by aerosol liquid water
- content, relative humidity, and oxidants.

## 42 1. Introduction

- Organic aerosol (OA) constitutes a significant component of PM<sub>2.5</sub>, accounting for 20%~80%
- of PM<sub>2.5</sub> mass (He et al., 2020; Zhang et al., 2007). Biomass burning (BB) is one of the primary
- sources of OA in the atmosphere and has significant impacts on air quality, visibility, and climate
- (Bao et al., 2021; Liu et al., 2019). BB releases various organic compounds, such as anhydro-
- saccharides, polycyclic aromatic hydrocarbons, and n-alkanes, with levoglucosan generally being
- the most abundant anhydro-saccharides (Chen et al., 2017; Yan et al., 2019). Levoglucosan and its
- two isomers, mannosan and galactosan, which are produced during the pyrolysis of cellulose and
- hemicellulose at temperatures ranging from 150 to 350 °C (Fabbri et al., 2009; Hong et al., 2022;
- Stevens et al., 2024), have been widely used as molecular markers of BB aerosol in PM<sub>2.5</sub> source
- apportionment (Alvi et al., 2020; Cheng et al., 2022; Hong et al., 2022; Kang et al., 2018; Li et al.,
- 2021; Liang et al., 2016).
- For decades, the anhydro-saccharides are considered as stable compounds and the majority of
- previous studies did not consider the degradation of anhydro-saccharides during the source
- apportionment of PM<sub>2.5</sub>. However, researchers found that levoglucosan can undergo oxidation in
- both the gas phase and liquid phase, or undergo heterogeneous oxidation on the aerosol surface (Bai
- et al., 2013; Li et al., 2021; Zhao et al., 2014). Recent studies have proved that the contribution of

https://doi.org/10.5194/egusphere-2025-5481 Preprint. Discussion started: 18 November 2025 © Author(s) 2025. CC BY 4.0 License.

63

67

70

74

8485

saccharides was ignored (Hong et al., 2022; Li et al., 2023a). For example, Li et al. (2023a) found that approximately 87% of levoglucosan had already degraded before reaching the receptor site, causing a 14.9 % underestimation of BB-derived OC. A recent study (Wang et al., 2025) in Hong Kong calculated the degradation rate constants and estimated the atmospheric lifetime of levoglucosan, but the universal of this phenomenon, and their driving factors that influence this degradation process have not been thoroughly investigated (Wang et al., 2025). Furthermore, it is currently unclear how the variations of atmospheric conditions influence the degradation rate of anhydro-saccharides. Traditional studies on levoglucosan degradation typically rely on offline sampling, laboratory experiments, and model simulations (Arangio et al., 2015; Bai et al., 2013; Hennigan et al., 2010; Liu et al., 2019; Xu et al., 2020). However, low temporal resolution observation limits the in-depth understanding of its degradation process, and discrepancies between laboratory experiments and theoretical studies can introduce large biases in model simulations. With the development of the Thermal Desorption Aerosol Gas Chromatography-Mass Spectrometry (TAG-GC/MS) system, it is now possible to collect organic marker data with high temporal resolution, capturing dynamic changes in emission sources and the aging process of organic aerosols (He et al., 2020; Li et al., 2020; Wang et al., 2020; Zhang et al., 2021a; Zhao et al., 2013; Zhu et al., 2021). Therefore, online observations across multiple cities under different atmospheric environments are essential to better understand the degradation rates of anhydro-saccharides. Zibo, Changzhou, and Hong Kong are three representative cities located in the North China Plain (NCP), the Yangtze River Delta (YRD) region, and the Pearl River Delta (PRD) region, respectively. These cities exhibit significant disparities in meteorological conditions, anthropogenic sources, and ambient air pollution levels. These divergent environmental conditions imply that the degradation process of anhydro-saccharides in the atmosphere may be influenced by different factors. To investigate this, a comparative analysis of anhydro-saccharides degradation rates in these cities was conducted. The bihourly resolution data of PM2.5-bound levoglucosan, mannosan and galactosan were collected at Zibo, Changzhou and Hong Kong during cold season using TAG-GC/MS system. The daytime decay rates of anhydro-saccharides in three cities were calculated using the relative rate constant method, and the driving factors influencing the daytime decay rates of anhydro-saccharides were analyzed with the generalized additive model (GAM). The findings of

BB to organic carbon (OC) and PM<sub>2.5</sub> could be underestimated if the degradation of anhydro-

- this study enhance our understanding of atmospheric degradation mechanisms of anhydro-
- saccharides and provide a scientific foundation for precise source apportionment, especially for
- refining our knowledge on the source contributions to PM<sub>2.5</sub> from biomass burning.

## 2. Methodology

#### 2.1 Site description and field observation

Field observations using TAG-GC/MS were conducted during autumn and winter season across three typical cities in the three regions (Zibo, Shandong province, NCP; Changzhou, Jiangsu province, YRD; and Hong Kong, PRD region) over eastern China (Fig. S1). The sampling site in Zibo was situated at the Zibo Ecological Environment Monitoring Station (36°50'N, 118°07'E). This location is bordered by Lutai Avenue approximately 500 m to the east, Lushan Avenue approximately 100 m to the south, and residential neighborhoods approximately 1 km to the north. It represents the urban atmospheric environment influenced by multiple pollution sources including both anthropogenic and biogenic emissions. The vegetation in the vicinity mainly consists of temperate deciduous broadleaf forests. Organic compounds emitted from this area may significantly impact ground-level aerosols. The observation period was from November 2022 to February, 2023. The sampling site in Changzhou was situated at the Changzhou Environmental Monitoring Center (119°59.730'E, 31°45.510'N). The surrounding environment includes numerous commercial and residential districts, as well as major roads such as Zhongwu Avenue, Heping Middle Road, and Guanghua Road, representing an urban environment affected by various pollution sources. Data from January to March 2021 was collected, and the detailed information of this filed campaign can be found in our previous studies (Li et al., 2023b; Yi et al., 2024). The sampling site in Hong Kong was at the Hong Kong University of Science and Technology Super Station (114°16'E, 22°19'N), situated in a suburban area with relatively limited local emissions. Observations were conducted from October 2020 to January 2021. During the observation periods, the anhydro-saccharides data had a temporal resolution of 2 hours; however, some data were lost due to instrument malfunctions or maintenance during certain intervals. During the field campaign, we conducted online measurements at the three stations to collect auxiliary data, including meteorological conditions, air pollutant and PM2.5 concentrations. At

Changzhou site, meteorological parameters including wind speed (WS), wind direction (WD),

WXT520 (VAISALA, FL); PM2.5 by BAM1020 (Met One, US) via beta-ray method; O3, NOx by 120 MODEL 49i, MODEL450i (Thermo Fisher Scientific, US) respectively; OC/EC by RT-4 (Sunset 121 Laboratory, US) and water-soluble ions (Cl<sup>-</sup>, NO<sub>3</sub><sup>-</sup>, SO<sub>4</sub><sup>2-</sup>, Na<sup>+</sup>, NH<sub>4</sub><sup>+</sup>, K<sup>+</sup>, Mg<sup>2+</sup>, Ca<sup>2+</sup>) by ADI2080 122 (Metrohm, CHN). The Solar surface radiation data (SSR) were obtained from the ERA5-Reanalysis 123 (https://cds.climate.copernicus.eu/datasets). In Zibo, meteorological data including wind speed 124 (WS), wind direction (WD), relative humidity (RH), temperature (T), atmospheric pressure (P), 125 rainfall (RF) were obtained from the China Meteorological Administration 126 (https://www.cma.gov.cn/); PM2.5 by MODEL 5014i (Thermo Fisher Scientific, US); O3, NO<sub>x</sub> by 127 MODEL 49i, MODEL 42i (Thermo Fisher Scientific, US); OC/EC by MODEL ECOC-610 128 (Hangzhou Pengpu Technology Co., Ltd., China), water-soluble ions (Cl<sup>-</sup>, NO<sub>3</sub><sup>-</sup>, SO<sub>4</sub><sup>2-</sup>, Na<sup>+</sup>, NH<sub>4</sub><sup>+</sup>, 129 K<sup>+</sup>, Mg<sup>2+</sup>, Ca<sup>2+</sup>) by MODEL S611 (Fortelice International Co., Ltd., Taiwan, China) and solar surface 130 radiation (SSR) by CMP11 (Kipp & Zonen, Zuid-Holland, Netherlands). At Hong Kong site, PM2.5 131 was measured by Model 5030i (Thermo Fisher Scientific, US); water-soluble ions (Cl<sup>-</sup>, NO<sub>3</sub><sup>-</sup>, SO<sub>4</sub><sup>2-</sup>, Na+, NH<sub>4</sub>+, K+, Mg<sup>2+</sup>, Ca<sup>2+</sup>) and OC/EC by ADI2080 (Metrohm, CHN) and RT-4 (Sunset Laboratory, 132 133 US) respectively; meteorological parameters wind speed (WS), wind direction (WD) relative 134 humidity (RH), temperature (T), atmospheric pressure (P), rainfall (RF), O<sub>3</sub>, NO<sub>x</sub> and SSR by AWS 135 tower (Hong Kong Environment Protection Department); elemental species (K, Ca) by Xact 625i 136 (Cooper Environmental Services) via X-ray method. Detailed information about online observations 137 can be found in Table S1. 138 The TAG-GC/MS system is applied for the online measurement of levoglucosan, galactosan 139 and mannosan. Detailed description and the schematic diagram of TAG can be found in our previous 140 studies (He et al., 2020; Li et al., 2020; Wang et al., 2020; Wang et al., 2025; Zhang et al., 2021a). 141 During the observation, deuterium-labeled internal standard solution was injected into each sample 142 to monitor instrument condition and analyze the contamination levels of key species. The detailed description is provided in Text S1. The identification of the target saccharides is achieved by 143 144 comparing their retention times and mass spectra with those of authentic standards. Subsequently, 145 quantitative analysis is performed using internal standard calibration. We plotted the relationship 146 between the peak area ratios of external standard solutions and the concentrations of target 147 compounds in the standard mixture to generate a calibration curve, and the correlation coefficients

relative humidity (RH), temperature (T), atmospheric pressure (P), rainfall (RF) were monitored by

- 148 (R) ranges from 0.92 to 0.99. The detailed information on the preparation of external standard 149 solutions and internal standard solutions for the saccharide compounds analyzed in the field
- campaign can be found in Table S2 and Table S3.

#### 2.2 ISORROPIA-II Model

- Aerosol acidity (pH<sub>is</sub>) and aerosol liquid water content (ALWC) were calculated using the
- forward mode of the ISORROPIA-II model (http://isorropia.eas.gatech.edu)(C. Fountoukis and
- Nenes, 2007). The input parameters required for the model primarily included water-soluble
- inorganic ions (SO<sub>4</sub><sup>2</sup>, NO<sub>3</sub>, NH<sub>4</sub><sup>+</sup>, K<sup>+</sup>, Ca<sup>2+</sup>, Na<sup>+</sup>, Mg<sup>2+</sup>, and Cl<sup>-</sup>), NH<sub>3</sub>, temperature (T), and relative
- humidity (Hennigan et al., 2015). The calculation formula is as follows:

$$pH_{ls} = -lg(\frac{1000 \times H^{\perp}}{ALWC}) \tag{1}$$

- where H<sup>+</sup> represents the liquid-phase mass concentration of hydrogen ions, expressed in μg/m<sup>3</sup>,
- ALWC denotes the aerosol liquid water content, expressed in μg/m<sup>3</sup>.

## 160 2.3 Relative Rate Constant Method

- Previous studies have indicated that K<sup>+</sup> can serve as a tracer for biomass burning (Hong et al.,
- 2022; Pio et al., 2008). Moreover, the ratio of levoglucosan to BB-derived K<sup>+</sup> (K<sup>+</sup><sub>BB</sub>) observed in
- the environment can be used to distinguish different types of biomass burning, such as crop residue
- burning and wood combustion (Cheng et al., 2013). Additionally, due to the chemical stability of
- K<sup>+</sup> in the atmosphere, this ratio can also serve as an indicator of BB aerosol aging. However, since
- potassium ions can also originate from sea salt and dust (Karavoltsos et al., 2020; White, 2008),
- $K_{BB}^{+}$  should be calculated first.  $K_{BB}^{+}$  can be obtained by Equations (2)~(7):

$$K^{+}_{BB} = (K^{+}_{Nss}) - (K^{+}_{Dust})$$
 (2)

$$K^{+}_{Nss} = K^{+}_{aerosol} - 0.037*Na^{+}_{aerosol}$$
 (3)

$$K^{+}_{Dust} = 0.04 * [(Ca^{2+}_{Nss}) - Ca^{2+}_{BB}]$$
 (4)

$$Ca^{2+}{}_{BB} = K^{+}{}_{NSS}/2 \tag{5}$$

$$Ca^{2+}_{Nss} = Ca^{2+}_{aerosol} - 0.038*Na^{+}_{aerosol}$$
 (6)

$$z = (K^{+}_{NSS}/Ca^{2+}_{NSS})_{max} - (K^{+}_{NSS}/Ca^{2+}_{NSS})_{min}$$
 (7)

In Equations (2), K<sup>+</sup><sub>Nss</sub> and K<sup>+</sup><sub>Dust</sub> refer to non-sea-salt potassium and potassium originating from dust, respectively. In Equation (3), K<sup>+</sup><sub>aerosol</sub>, Na<sup>+</sup><sub>aerosol</sub>, and Ca<sup>2+</sup><sub>aerosol</sub> represent the concentrations of potassium, sodium, and calcium in the aerosol samples, which are the measured

204205

values. Based on previous literature, the mass ratios of (K+/Na+) and (Ca2+/Na+) in seawater are 0.037 and 0.038, respectively, and are used for the correction of sea salt aerosols (Kumar et al., 2018; Pio et al., 2007). The study of Kumar V et al(2018) suggests that the maximum and minimum differences in the mass ratio of  $(K_{Nss}^+/Ca^{2+}_{Nss})$  are considered to represent emissions from biomass burning (Kumar et al., 2018; Pio et al., 2008; Pio et al., 2007). The Ca<sup>2+</sup> originating from biomass burning is calculated by using (K<sup>+</sup><sub>Nss</sub>/Ca<sup>2+</sup><sub>Nss</sub>)<sub>max</sub> minus (K<sup>+</sup><sub>Nss</sub>/Ca<sup>2+</sup><sub>Nss</sub>)<sub>min</sub> as the denominator. In this study, the calculation method for the anhydro-saccharides decay rate was adopted from Wang et al. (2025), which is a variant of the relative rate constant approach utilizing inert K<sup>+</sup><sub>BB</sub> as the reference substance (Donahue et al., 2005; Wang et al., 2025). The validity of this method has been demonstrated in previous studies (Wang and Yu, 2021). The detailed derivation of the formula can be found in Text S2. The final derived expression is presented as Equation (8). According to (Wang et al., 2025), the C<sub>i</sub> represents the particle phase concentration of anhydro-saccharides, C<sub>K</sub><sup>+</sup><sub>BB</sub> represents the concentration of K<sup>+</sup> from biomass combustion, k<sub>2</sub> represents the second-order reaction rate constant between anhydro-saccharides and oxidants, and C<sub>OX</sub> represents the average concentration of oxidants in the aerosol phase. The calculated k corresponds to the effective total decay rate of anhydro-saccharides, which results from various atmospheric processes (such as heterogeneous oxidation and aqueous-phase oxidation). We assume that the emission of anhydrosaccharides and K<sup>+</sup><sub>BB</sub> is equivalent, or that no new pollutants are emitted, or that such emissions are negligible within the time frame of the study (i.e., 8 hours). Compared to previous studies applying the relative rate constant method to a pair of target and unknown compounds (Donahue et al., 2005; Huff Hartz et al., 2007), this method can be regarded as a special case where the reference species is inert and zero-corrected.  $\frac{\partial \ln (C_i/C_{K^+BB})}{\partial t} = -k, k = k_2 \times C_{OX}$ (8)

#### 2.4 Generalized Additive Models

Generalized Additive Models (GAM) are used to construct nonlinear regression relationships between explanatory variables and response variables. Unlike statistical distribution-based models, GAM is primarily data-driven, allowing for flexible adjustment of the functional form of the response variable based on the specific context (Charles and J.Stone, 1985). Compared to other statistical models, GAM offers higher flexibility and degrees of freedom, which does not require a

pre-defined parametric model, and can be applied to various distribution types, and can directly handle complex nonlinear relationships between explanatory and response variables (Zhai et al., 2019). The GAM model has been widely applied in studies investigating the influencing factors of nonlinear atmospheric pollutants, such as PM<sub>2.5</sub>, O<sub>3</sub>, and SOA(Hu et al., 2022; Zhang et al., 2021b). The basic form of the GAM model is shown in Equation 9.

$$g(\mu) = f_1(X_1) + f_2(X_2) + \dots + f_n(X_n) + \beta$$
 (9)

In the equation,  $g(\mu)$  is a continuous function representing the relationship between the nonlinear formula and the expected value;  $\mu$  denotes the response variable, i.e., the mass concentration of the target substance;  $\beta$  is the intercept;  $f_n$  (n=1, 2, ..., n) is the smoothing function connecting the explanatory variables;  $X_n$  (n=1, 2, ..., n) refers to the different explanatory variables. The significance of the explanatory variables is tested using the Akaike information criterion, and the most appropriate  $g(\mu)$  and  $X_n$  are selected through multiple linear tests. The  $R^2$ , deviance explained (%), and p-value calculated from the GAM model are used to assess the significance level, importance of  $X_n$ , and the model's goodness of fit.

## 3. Results and Discussions

#### 3.1 Overview of the field campaign

The overall environmental condition and air pollutant levels during the three field campaigns are summarized in Table S4. During the field observation period,  $PM_{2.5}$  pollution was the most severe in Zibo, with an average concentration of  $69.4 \pm 58.0 \,\mu\text{g/m}^3$ . In comparison, the average  $PM_{2.5}$  concentrations in Changzhou and Hong Kong were  $49.9 \pm 26.4 \,\mu\text{g/m}^3$  and  $31.5 \pm 20.5 \,\mu\text{g/m}^3$ , respectively. Zibo is a traditional heavy industrial city with more than 8,200 industrial plants, including power plants, chemical factories, and building materials industries (Pan et al., 2025). Due to its high emission intensity, Zibo has been suffering from poor air quality, consistently ranking last in air quality within Shandong Province in recent years. In contrast, the air quality in Hong Kong was relatively good, which is likely related to relatively low emission intensity and favorable meteorological conditions. The average winter temperature in Zibo (-0.2  $\pm$  6.1 °C) was the lowest among the three cities, with a maximum wind speed of  $2.2 \pm 1.7 \,\text{m/s}$ . As a city in northern China, Zibo's cold and windy climate is closely associated with the influence of northern continental air masses during winter. In contrast, Changzhou ( $10.9 \pm 4.9 \,^{\circ}\text{C}$ ,  $1.3 \pm 0.7 \,\text{m/s}$ ) and Hong Kong ( $15.6 \,^{\circ}$ 

$\pm$  10.5 °C, 1.6  $\pm$  0.5 m/s), located in southern and coastal regions, are influenced by the subtropical 236 monsoon and oceanic climate, leading to significantly higher winter temperatures and lower wind 237 speeds, resulting in a milder and more stable climate. As shown in Fig. 1, levoglucosan is the 238 dominant anhydrosugar in all three cities, with concentrations of  $45.5 \pm 32.3$  ng/m³ in Zibo,  $45.1 \pm$ 239 38.7 ng/m<sup>3</sup> in Changzhou, and 27.5 ± 15.6 ng/m<sup>3</sup> in Hong Kong, respectively. The average 240 concentration of mannosan in Changzhou  $(3.6 \pm 3.2 \text{ ng/m}^3)$  is higher than in Zibo  $(2.4 \pm 1.7 \text{ ng/m}^3)$ 241 and Hong Kong (1.9  $\pm$  1.5 ng/m³). Conversely, the concentration of galactosan in Zibo (4.5  $\pm$  3.4 242  $ng/m^3$ ) is significantly higher than in Changzhou (2.4 ± 2.0  $ng/m^3$ ) and Hong Kong (0.9 ± 0.7  $ng/m^3$ ). 243 These differences may reflect varying BB source types across the cities. The total concentration of 244 three anhydrosugars (levoglucosan, mannosan, and galactosan) in these three cities accounts for 0.7% 245 (Zibo), 0.9% (Changzhou), and 0.8% (Hong Kong) of the measured total organic carbon (OC) mass. 246 Although their proportion is relatively low, these anhydrosugars, as characteristic tracers of biomass 247 burning (BB), can be used to inversely estimate the contribution of BB sources to atmospheric OC. 248 Thus, they are key indicators for quantifying the impact of BB source emissions (Cheng et al., 2022; 249 Fabbri et al., 2009; Li et al., 2023a). 250 By analyzing the ratio of levoglucosan (Lev) to mannosan (Man), different types of biomass 251 combustion sources can be identified. For example, previous studies have shown that the Lev/Man 252 ratio from crop straw combustion can exceed 40, while the Lev/Man ratios from hardwood and 253 softwood combustion range from 15 to 25 and 3 to 10, respectively (Engling et al., 2009; Fu et al., 254 2012; Sang et al., 2013; Xu et al., 2020). We calculated the Lev/Man and Man/Gal ratios for the 255 three cities, as shown in Fig. 1. The average Lev/Man ratios in Zibo, Changzhou, and Hong Kong 256 were  $19.3 \pm 5.6$  (range:  $5.2 \sim 43.3$ ),  $13.9 \pm 6.6$  (0.9 $\sim 56.8$ ), and  $17.8 \pm 6.5$  (5.6 $\sim 51.4$ ), respectively. 257 Fig. S2 illustrates the parameter ratio space of Lev/Man and Lev/K<sup>+</sup> to characterize biomass burning 258 characteristics and distinguish different combustion types. The tracer ratio space diagram for 259 coniferous trees, deciduous trees, hardwood, softwood, and crop residues, proposed by Cheng et al. 260 (2013), overcomes the limitation of relying solely on a single feature ratio (such as Lev/K<sup>+</sup> or 261 Lev/Man) for distinguishing biomass burning types (Cheng et al., 2013). It is important to note that, 262 due to insufficient observational data for K<sup>+</sup> in Hong Kong, while the total potassium (total K) data 263 is relatively complete, total potassium (KBB) was used as a substitute for K+ in the subsequent 264 combustion type diagnosis. Detailed information can be found in our previous study (Wang et al.,

https://doi.org/10.5194/egusphere-2025-5481 Preprint. Discussion started: 18 November 2025 © Author(s) 2025. CC BY 4.0 License.

265

267268

277278

2025). The ratio ranges for Zibo, Changzhou, and Hong Kong all fall within the range typically associated with crop residue burning, which is consistent with previous studies. The study period coincided with the autumn and winter seasons, which correspond to the typical period of crop residue burning (Cheng et al., 2013; Wang et al., 2020). In addition, the ratio of mannosan to galactosan (Man/Gal) has been used as an auxiliary method for distinguishing biomass combustion sources. During the observation period, the average Man/Gal values for Changzhou and Hong Kong were  $1.56 \pm 0.75$  (range:  $0.32 \sim 13.81$ ) and  $2.30 \pm 0.64$  (range:  $0.84 \sim 6.38$ ), respectively, consistent with previous studies which indicate that in the combustion emissions of crop straw, grass, and coal pellets, the content of mannosan (Man) is usually higher than that of galactosan (Gal) (Fabbri et al., 2009; Vicente et al., 2018; Xu et al., 2020). However, the average Man/Gal value in Zibo was 0.56 ± 0.19 (range: 0.23~2.24), significantly lower than those in Changzhou and Hong Kong, with a relatively higher concentration of galactosan, which may be related to differences in the type of combustion source or combustion conditions (Haque et al., 2022; Kuo et al., 2011; Yan et al., 2018). For example, the combustion of coal and certain industrial fuels may lead to higher galactosan content due to differences in the organic composition of these fuels compared to biomass fuels (Yan et al., 2018). Furthermore, incomplete combustion or low-temperature combustion may increase galactosan concentration (Haque et al., 2022), which could be a characteristic feature of combustion in Zibo. As a heavy industrial city, Zibo have more industrial combustion sources and incomplete combustion phenomena, leading to the relative enrichment of galactosan and exhibiting distinct chemical characteristics compared to common biomass combustion.

Fig. 1 Boxplot of the concentrations of levoglucosan (Lev), mannosan (Man), galactosan (Gal), and their ratios (Lev/Man, Man/Gal) in the cities of Zibo, Changzhou, and Hong Kong.

# 

Fig. 2 shows the detailed diurnal variations of anhydro-saccharides across three cities. The anhydro-saccharides in all three cities exhibit similar diurnal variation characteristics, indicating the similar source and atmospheric degradation processes. Specifically, the concentrations of anhydro-saccharides in the three cities generally decrease during the daytime (8:00-16:00 LST) and increase at night. Notably, in Fig. 2(b), the decline of levoglucosan in Changzhou is most pronounced during the daytime. In Changzhou, daytime levoglucosan concentrations peak at 08:00 LST ( $49.26 \pm 5.71$  ng/m³) before gradually declining to a minimum of  $29.53 \pm 4.54$  ng/m³ by 14:00 LST. A slight rebound to  $30.56 \pm 4.56$  ng/m³ occurs at 16:00 LST, followed by a significant increase to a secondary peak of  $59.88 \pm 5.71$  ng/m³ at 20:00 LST. In addition, Fig. 2(c) show the diurnal variation of levoglucosan in Hong Kong exhibits a pronounced pattern, with peak concentrations occurring in the morning at 8:00 LST, ( $32.65 \pm 1.72$  ng/m³). Similar to Changzhou, the levoglucosan concentration drops to a minimum at 14:00 LST ( $22.40 \pm 1.24$  ng/m³), followed by a slight rebound to  $23.05 \pm 1.20$  ng/m³ at 16:00 LST. The diurnal variation of levoglucosan in Zibo is relatively flat, peaking at 8:00 LST ( $48.24 \pm 4.29$  ng/m³) and declining to  $39.83 \pm 3.30$  ng/m³ at 16:00 LST. Similarly, the concentration of mannosan in Changzhou shows the most significant daytime

308309

decrease, from a peak at  $8:00 (3.89 \pm 0.40 \text{ ng/m}^3)$  to a minimum at  $16:00 (2.42 \pm 0.45 \text{ ng/m}^3)$ . A significant decline was also observed in Hong Kong, from  $2.42 \pm 0.20$  ng/m<sup>3</sup> at 8.00 to  $1.47 \pm 0.10$ ng/m<sup>3</sup> at 16:00 LST. In contrast, the diurnal variation of mannosan in Zibo is relatively smooth, with an initial concentration of  $2.38 \pm 0.19$  ng/m³ at 08:00 LST, and decreases to  $2.28\pm0.18$  ng/m³ at 16:00 LST. Galactosan in all three cities exhibited a congruent diurnal trend to the other anhydrosaccharides, characterized by a daytime decrease. For instance, galactosan concentrations in Zibo declined from a peak of  $4.60 \pm 0.38$  ng/m<sup>3</sup> at 8.00 LST to  $4.00 \pm 0.33$  ng/m<sup>3</sup> at 16.00 LST. In Changzhou, the galactosan concentration was  $2.65 \pm 2.39$  ng/m<sup>3</sup> at 8.00 LST, and it dropped to 1.79 $\pm 0.30$  ng/m<sup>3</sup> at 16:00 LST. In Hong Kong, the galactosan concentration decreased from  $1.08 \pm 0.09$  $ng/m^3$  at 8:00 LST was to  $0.67 \pm 0.06$   $ng/m^3$  by 16:00 LST. The concentration and compositional characteristics of organic aerosols in real atmospheric environments are governed by a combination of factors, including the intensity of air pollutant source emissions, variations in meteorological conditions, atmospheric chemical reactions, atmospheric diffusion capabilities, and deposition processes. These interrelated influences collectively impart high complexity and uncertainty to both the concentration levels and chemical composition of organic aerosols (Chen et al., 2022; Kim et al., 2017; Zhang et al., 2013). Previous studies have indicated that the ratio of levoglucosan to potassium ions (K<sup>+</sup>) from BB sources (levoglucosan/K<sup>+</sup><sub>BB</sub>) is an effective indicator of the aging degree of BB aerosols(Cheng et al., 2013; Li et al., 2021; Mochida et al., 2010). Therefore, to investigate the diurnal decay pattern of anhydro-saccharides, we selected levoglucosan/K+BB to examine the loss of levoglucosan. It is important to note that due to the insufficient observational data for K<sup>+</sup> in Hong Kong, and the relatively complete total potassium (total K) data, total potassium (K<sub>BB</sub>) was used as a substitute for K<sup>+</sup><sub>BB</sub> in Hong Kong. The analysis revealed a good correlation between the calculated  $K_{BB}$  and levoglucosan, with  $R_p = 0.63$ . Detailed information can be found in our previously published paper (Wang et al., 2025). In Zibo and Changzhou, the calculation formulas for  $K_{BB}^+$  were (2)-(7); correlation analysis results showed a significant positive correlation between levoglucosan and K+BB across different sites (as shown in Fig. S3). In Zibo, the pearson correlation coefficients between levoglucosan, mannosan, and galactosan and K<sup>+</sup><sub>BB</sub> were 0.65, 0.52, and 0.52, respectively. In Changzhou, the pearson correlation coefficients between levoglucosan, mannosan, and galactosan and K+BB were 0.76, 0.58, and 0.54, respectively, further confirming the validity and reliability of the K<sup>+</sup><sub>BB</sub> calculation formulas for Zibo and Changzhou.

Fig. 2 Diurnal variations of levoglucosan, mannosan, and galactosan; Lev/ $K^+_{BB}$ , Man/ $K^+_{BB}$ , and Gal/ $K^+_{BB}$  ratios at (a) Zibo, (b) Changzhou and (c) Hong Kong.

## 3.3 Daytime Decay Rate Calculation

Using equation (8), we calculated the daytime decay rates of anhydro-saccharides in three cities. The decay rate of levoglucosan in Changzhou was  $0.13 \pm 0.05 \text{ h}^{-1}$  (range:  $0.01 \sim 0.55 \text{ h}^{-1}$ ), ranking first among the three cities, indicating a significant decay rate. The decay rates of levoglucosan in Zibo and Hong Kong were similar, with values of  $0.10 \pm 0.08 \text{ h}^{-1}$  (range:  $0.001 \sim 0.34 \text{ h}^{-1}$ ) and  $0.10 \pm 0.05 \text{ h}^{-1}$  (range:  $0.02 \sim 0.25 \text{ h}^{-1}$ ), respectively. However, Hong Kong had the fastest decay rates for the other two anhydro-saccharides, especially galactosan, with a decay rate of  $0.15 \pm 0.06 \text{ h}^{-1}$  (range:  $0.04 \sim 0.33 \text{ h}^{-1}$ ), much higher than the other cities. In comparison, Changzhou's galactosan decay rate was  $0.13 \pm 0.08 \text{ h}^{-1}$  (range:  $0.01 \sim 0.67 \text{ h}^{-1}$ ), and Zibo's galactosan decay rate was  $0.10 \pm 0.03 \text{ h}^{-1}$  (range:  $0.004 \sim 0.31 \text{ h}^{-1}$ ), both of which were relatively smaller. Similarly, the decay rate of mannosan

in Hong Kong was the highest, reaching  $0.14 \pm 0.05 \ h^{-1}$  (range:  $0.04 \sim 0.29 \ h^{-1}$ ), followed by Changzhou at  $0.13 \pm 0.07 \ h^{-1}$  (range:  $0.01 \sim 0.60 \ h^{-1}$ ), while Zibo had the lowest rate at only  $0.09 \pm 0.03 \ h^{-1}$  (range:  $0.01 \sim 0.33 \ h^{-1}$ ). The detailed decay rates of anhydro-saccharides for the three cities are presented in Table S5. Based on the distribution of decay rates in three cities shown in Fig. 3, the average decay rate of levoglucosan in Changzhou is higher than in the other two cities, with 33.3% of the rates exceeding  $0.15 \ h^{-1}$ . In contrast, the proportions for Zibo and Hong Kong are 12.9% and 13.0%, respectively. Moreover, Hong Kong shows a remarkable distribution of decay rates for mannosan and galactosan, with 37.7% of mannosan decay rates and 37.7% of galactosan decay rates exceeding  $0.15 \ h^{-1}$ . These results indicate that in the three cities, Changzhou exhibits the highest average decay rate for levoglucosan, while Hong Kong shows the highest decay rates for mannosan and galactosan, and Zibo has the lowest decay rates for all three sugars.

Fig. 3 Distribution of the decay rates of levoglucosan, mannosan and galactosan for (a) Zibo, (b)

Changzhou and (c) Hong Kong.

It is noteworthy that not all sampling days showed good linear fitting, with some days having negative decay rates, indicating that some of the daytime data did not follow the decay trend. This may be due to the direct emission and transmission of BB air masses. Zibo sampled for a total of 66

days, with 31 days fitting the linear decay pattern; Changzhou sampled 42 days, with 21 days fitting the linear decay pattern; and Hong Kong sampled 106 days, with 69 days fitting the linear decay pattern. Fig. 4 shows examples of linear fitting in the three cities, with the x-axis representing five time points: 8:00, 10:00, 12:00, 14:00 and 16:00 each day, and the y-axis representing  $\ln(\frac{c_i}{C_{K_{RB}^+}})$ ,

with the slope equal to k.

Fig. 4 Example calculation of the degradation rates of levoglucosan, mannosan and galactosan at

(a) Zibo, (b) Changzhou and (c) Hong Kong.

The three anhydro-saccharides differ in molecular structure, particularly in the C-H bonds at different positions on the sugar rings. This causes a different potential in the way hydroxyl radicals (·OH) react with each sugar molecule. Nevertheless, all three sugars undergo oxidation by ·OH radicals. Therefore, we conducted a correlation analysis of the decay rates for the anhydro-saccharides in each city. As shown in Fig. S4, there is a strong correlation among the decay rates of anhydro-saccharides in the three cities. Although the molecular structures of the sugars differ, the oxidation mechanism by ·OH is similar, which explains the high correlation between their decay rates. Section 3.4 provides a detailed exploration of the environmental factors influencing the decay rate of anhydro-saccharides. According to John et al. (2020a and 2020b), the bound dissociation enthalpy (BDE) of levoglucosan, mannosan and galactosan can be estimated by the Accurate Bond dissociation Enthalpy Tool (ALFABET) online tool (<a href="https://bde.ml.nrel.gov/">https://bde.ml.nrel.gov/</a>, last access: 21 Oct 2025). The results show that the most easily broken C-H bonds for levoglucosan, mannosan, and

galactosan are all at positions 3 and 2, with corresponding BDE of 85.3~86.9 kcal/mol, 84.6~85.1 kcal/mol, and 82.9~84.6 kcal/mol, respectively. This implies that the decay rates of the three anhydro-saccharides should be galactosan > mannosan > levoglucosan (St John et al., 2020a; St John et al., 2020b). However, this rule was only found in Hong Kong and Changzhou. Hence, except for bond dissociation energies (BDEs), there should be other driving factors of the decay rate.

#### 3.4 Driving factors of the decreasing rate of anhydro-saccharides

The three sampling points represent cities with distinct meterological conditions. Zibo has a temperate monsoon climate, characterized by cold and dry winters. In contrast, Changzhou falls under the subtropical monsoon climate, with winters being more humid than those in Zibo. Although Hong Kong also has a subtropical monsoon climate, it is significantly influenced by the oceanic climate, resulting in smaller temperature variations and a more humid winter. The differences in climate conditions indirectly lead to variations in environmental factors, which in turn affect the daytime degradation rate of anhydro-saccharides across different cities. Therefore, we compared the environmental factors of the three cities with the calculated degradation rates of anhydro-saccharides. These factors include ALWC related to liquid-phase reactions, the atmospheric oxidative capacity indicator  $O_x$ , solar surface radiation (SSR), relative humidity (RH) and temperature (T). Due to the lack of data on gas-phase anhydro-saccharides, the gas-phase oxidation part was not discussed in this study.

As an indicator of the total amount of various oxidants in the atmosphere,  $O_x$  is used in this study to explore its impact on the daytime degradation rate of levoglucosan. As shown in Fig. 5, the  $O_x$  concentration in Changzhou is higher than in the other two cities, with an average value of 88.9  $\pm$  26.0  $\mu$ g/m³ (range: 41.4~162.0  $\mu$ g/m³). In contrast,  $O_x$  levels in Zibo (79.4±16.3  $\mu$ g/m³) is slightly lower than Changzhou but much higher than that in Hong Kong (62.3  $\pm$  12.4  $\mu$ g/m³). D. Hoffmann et al. (2010) reported that the reaction of levoglucosan with ·OH is a major degradation pathway, and model calculations indicate that levoglucosan is more readily oxidized by ·OH during the daytime. The average degradation flux during the winter daytime is 4.7  $\mu$ g·m³·h¹-1(Hennigan et al., 2010). On the other hand, solar surface radiation (SSR), the key parameter for daytime ·OH production, showed similar levels in Changzhou and Hong Kong. The average solar radiation in Changzhou is 381.7  $\pm$  165.6 W/m² (range: 147.3~779.0 W/m²), while Hong Kong records 354.0  $\pm$  108.1 W/m² (range: 133.7~547.0 W/m²). Zibo, by contrast, is markedly lower at 273.6  $\pm$  81.8 W/m²

418

421

437

(range: 79.8~427.1 W/m<sup>2</sup>). Slade and Knopf (2014) reported that the presence of water can reduce particle viscosity, thereby enhancing 'OH oxidation (Slade and Knopf, 2014). Therefore, the higher degradation rate of levoglucosan in Changzhou compared to the other two cities may reflect stronger ·OH oxidation. As illustrated in Fig. 5(c), Changzhou exhibits the highest ALWC among the three cities, averaging  $15.6 \pm 15.5 \,\mu\text{g/m}^3$  (range:  $0.4 \sim 56.5 \,\mu\text{g/m}^3$ ). Zibo follows closely at 13.4  $\pm$  15.4 µg/m<sup>3</sup> (range: 0.7~55.7 µg/m<sup>3</sup>), while Hong Kong records the lowest levels at  $6.4 \pm 8.9 \mu g/m^3$ (range: 0.2~39.7 μg/m³). Furthermore, Slade and Knopf (2014) noted that an increase in relative humidity accelerates the heterogeneous oxidation rate of levoglucosan (Slade and Knopf, 2014). The relative humidity in Hong Kong is significantly higher than in the other two cities, with an average of 52.8 ± 13.0% (range: 15.8~88.0%). Zibo has the lowest relative humidity, with an average of 41.2 ± 16.4% (range: 18.4~89.8%), while Changzhou has an average relative humidity of  $48.9 \pm 19.2\%$  (range:  $20.4 \sim 80.2\%$ ). When the relationship between the response variable and explanatory variables is unclear, the generalized additive model (GAM) can be used to fit the explanatory and response variables by plotting smooth functions, further assessing their linear or nonlinear relationship (ref). For levoglucosan, the daytime degradation rate calculated for the three cities was used as the response variable in the GAM model, and the various influencing factors (Ox, ALWC, SSR, RH and T) were used as the corresponding explanatory variables. Additionally, the model's accuracy was assessed by examining the R<sup>2</sup>, p-values, and deviance explained (DE).

Fig. 5 Comparison of environmental factors and decay rates across three cities: (a) decay rates of levoglucosan, mannosan and galactosan, (b) O<sub>x</sub>, (c) ALWC, (d) SSR, (e) RH, (f)T.

The validation results of the GAM model are shown in Fig. S5. From the residual Q-Q plot (Fig. S5a), it can be observed that most of the data points approximately follow a straight line, indicating that the residuals of the GAM model generally follow a normal distribution. Fig. S5b

residuals are randomly distributed. The residual histogram (Fig. S5c) demonstrates a rough 441 symmetric distribution. Additionally, Fig. S5d reveals that the observed values closely follow the 442 "1:1" line, indicating a good fit between the observed and fitted values. These results confirm that 443 the GAM model provides a good fit for the daytime attenuation rate of levoglucosan. 444 The analysis results from the GAM model are shown in Table S6. The adjusted R<sup>2</sup> between the 445 observed and estimated values was 0.70, with a bias explanation rate of 70.9%. The daytime decay rate of levoglucosan significantly increased with the rise in ALWC (p<0.05). As shown in Fig. 6(a), 446 447 especially ALWC >30 μg/m<sup>3</sup>, the decay rate increased as ALWC increased, indicating that ALWC 448 is an important factor affecting the daytime decay rate. This result is consistent with the findings of 449 Slade and Knopf (2014), which suggest that liquid-phase reactions reduce the viscosity of 450 levoglucosan particles, allowing for easier absorption of OH and accelerating the daytime decay 451 process(Slade and Knopf, 2014). We also find that the decay rate significantly increased with 452 temperature (T) (p<0.05). However, temperature exerts a modest positive contribution to the decay 453 rate when  $T < 10^{\circ}$ C. In contrast, when  $T > 10^{\circ}$ C, the decay rate increases markedly, indicating that 454 the rise in temperature affects the decay rate. Additionally, the decay rate significantly increased 455 with the rise of  $O_x$  (p<0.05). As shown in Fig. 6(a), the decay rate almost increased linearly with  $O_x$ . 456 When the Ox>150 µg/m<sup>3</sup>, the decay rate is higher than 0.1 h<sup>-1</sup>, suggesting that oxidants become the 457 key driving factor for the attenuation reaction under this condition. This study also found that 458 relative humidity (p=0.08) and SSR (p=0.12) did not significantly affect the decay rate. However, 459 Fig. 6(a) show that both are positively correlated with the daytime decay rate of levoglucosan, 460 especially RH>60%, which nearly increases linearly. Although SSR did not show a significant 461 upward trend when it was less than 400 W/m<sup>2</sup>, when SSR exceeded 400 W/m<sup>2</sup>, the decay rate 462 increased with the rise in SSR, suggesting that SSR still has some effect on the decay rate. Further univariate GAM tests showed that, after excluding the interference of other variables, RH and SSR 463 464 were significantly positively correlated with the daytime decay rate of levoglucosan (p

similar characteristics to levoglucosan with respect to ALWC, RH,  $O_x$ , and SSR. As shown in Fig. 6(b) and Fig. 6(c), when temperature (T) 

 $Fig.\ 6\ Influence\ of\ various\ factors\ on\ the\ daytime\ degradation\ rate\ of\ levoglucos an\ analyzed$ 

#### using the GAM model. (a) ALWC, (b) T, (c) RH, (d) Ox, (e) SSR

As shown in Fig. 6, ALWC, RH, and Ox are the three factors that contribute most to the degradation rate of levoglucosan. Therefore, we plotted these factors against the degradation rate to better explore their relationship. As illustrated in Fig. 7, overall, all three factors show a positive correlation with the levoglucosan degradation rate. Specifically, as the concentration of Ox increases, relative humidity (RH) rises, and the liquid water content (ALWC) increases, the degradation rate of levoglucosan tends to increase. In contrast, when the levels of these three factors are relatively low, the degradation rate generally remains in a lower range, further confirming that these factors play a promoting role in the degradation process of levoglucosan.

 $Fig.\ 7\ Relationships\ between\ decay\ rates\ of\ levoglucosan\ (a),\ mannosan\ (b),\ and\ galactosan\ (c)$ 

with Ox, RH, and ALWC

Based on the results from the GAM model, we separately examined the contributions of five factors to the degradation rates of dehydrated sugars in Zibo, as shown in Fig. 8. The results indicate that  $O_x$  contributes more significantly to the degradation rate of levoglucosan than the other two anhydro-saccharides, while aerosol liquid water content (ALWC) contributes similarly to the degradation rates of mannosan and galactosan, which is significantly higher than that of levoglucosan. These results suggest that the degradation rate of dehydrated sugars is not only related to the bond dissociation energies (BDEs) of their structures but also influenced by other driving factors, with varying contributions. The combined effects of multiple driving factors and the differential sensitivities of different sugars to these factors ultimately lead to differences in the degradation rates. This may help explain why the degradation rates of the three dehydrated sugars in Zibo do not follow the pattern predicted by bond dissociation energies (BDEs).

Fig. 8 Contributions of ALWC, T, RH, and  $O_x$  to the decay rates of levoglucosan, mannosan, and galactosan in Zibo

In conclusion, the analysis results from the GAM model suggest that the daytime decay rate of

528529

anhydro-saccharides is influenced by multiple factors, with ALWC, T, and  $O_x$  being the main driving factors. Despite the lack of significance in the effects of RH and SSR, they still showed a positive correlation with the decay rate. Given that the sampling times at the three sampling sites in this study concentrated in the autumn and winter, it is expected that in summer, under conditions of higher  $O_x$ , temperature, and SSR, the decay rate of anhydro-saccharides will significantly increase. Therefore, further investigation into the degradation mechanisms of anhydro-saccharides is crucial for accurately assessing the contribution of BB aerosols to global air quality, particularly in the context of seasonal and environmental changes, and holds significant scientific and practical value.

## 4. Conclusions

This study employed TAG-GC/MS to obtain bihourly time resolution PM2.5-bound anhydrosaccharides (levoglucosan, mannosan, and galactosan) during the winter season in three typical cities over three regions in China, including Zibo, Changzhou and Hong Kong, located in the NCP, YRD and PRD regions, respectively. The decay rates of levoglucosan, mannosan, and galactosan in the real atmosphere and the driving factors are investigated. Results indicate that levoglucosan had the highest concentration among all the three anhydro-saccharides. In Zibo, the concentration of levoglucosan was  $45.5 \pm 32.3$  ng/m<sup>3</sup>, higher than Changzhou  $(45.1 \pm 38.7 \text{ ng/m}^3)$  and Hong Kong  $(27.5 \pm 15.6 \text{ ng/m}^3)$ . The diurnal variation of the three anhydro-saccharides showed a decreasing trend during the daytime (8:00-16:00). We selected  $K^{+}_{BB}$  as a reference substance and calculated the daytime degradation rates of the three anhydro-saccharides in the three cities using the relative rate constant method. The results indicated that the degradation rate of levoglucosan was highest in Changzhou, at  $0.13 \pm 0.05 \text{ h}^{-1}$  (range:  $0.01 \sim 0.55 \text{ h}^{-1}$ ), while mannosan and galactosan showed the highest degradation rates in Hong Kong, at  $0.14 \pm 0.05 \ h^{-1}$  (range:  $0.04 \sim 0.29 \ h^{-1}$ ) and  $0.15 \pm 0.06 \ h^{-1}$ <sup>1</sup> (range: 0.04~0.33 h<sup>-1</sup>), respectively. Due to structural differences, particularly the varying positions of the C-H bond within the sugar ring, the reaction mechanisms of hydroxyl radicals (OH) with the three sugar molecules differ, leading to variations in the degradation rates of anhydrous sugars within the same city. Environmental factors, such as air quality and climate type, in different cities further contribute to variations in degradation rates. In addition, the differential sensitivity of various anhydro-saccharides to these driving factors leads to differences in the decay rates of the three sugars. The GAM model results indicate that the daytime decay rate of anhydro-saccharides is

536 primarily influenced by ALWC, RH, and O<sub>x</sub>. Additionally, increases in T and SSR also contribute 537 to an enhanced decay rate. Our findings highlight that the degradation of anhydro-saccharides in real atmospheric conditions occurs through various oxidative mechanisms. Further investigation 538 539 into the degradation mechanisms of anhydro-saccharides is crucial for accurately assessing the 540 contribution of BB aerosols to global air quality. The results of this study provide valuable data and 541 insights for future air quality management. 542 Data availability. Data will be available upon request to the corresponding authors. 543 Author contributions. Conceptualization: Li Li, Yu Jianzhen, Kun Zhang; Formal analysis: Biao 544 Zhou, Kun Zhang, Jiqi Zhu; Methodology and Investigation: Biao Zhou, Kun Zhang, Qiongqiong 545 Wang; Writing - original draft: Biao Zhou; Writing - review & editing: Kun Zhang, Li Li, 546 Jianzhen Yu, Qiongqiong Wang; Funding acquisition and Supervision: Li Li. 547 Competing interests. The authors declare no conflicts of interest. 548 549 Acknowledgements. This study is financially supported by Jing-Jin-Ji Regional Integrated 550 Environmental Improvement - National Science and Technology Major Project (2025ZD1202005, 551 2025ZD1202004), the National Natural Science Foundation of China (42375102). This work is 552 supported by the Shanghai Technical Service Center of Science and Engineering Computing, 553 Shanghai University. 554 References 555 Alvi, M.U., Kistler, M., Shahid, I., Alam, K., Chishtie, F., Mahmud, T., Kasper-Giebl, A., 2020. 556 Composition and source apportionment of saccharides in aerosol particles from an agro-industrial zone 557 Indo-Gangetic Plain, Environ. Sci. Pollut. Res., 27, 14124-14137. 558 http://dx.doi.org/10.1007/s11356-020-07905-2. 559 Arangio, A.M., Slade, J.H., Berkemeier, T., Pöschl, U., Knopf, D.A., Shiraiwa, M., 2015. Multiphase 560 Chemical Kinetics of OH Radical Uptake by Molecular Organic Markers of Biomass Burning Aerosols: 561 Humidity and Temperature Dependence, Surface Reaction, and Bulk Diffusion, The Journal of Physical 562 Chemistry A, 119, 4533-4544. http://dx.doi.org/10.1021/jp510489z. 563 Bai, J., Sun, X., Zhang, C., Xu, Y., Qi, C., 2013. The OH-initiated atmospheric reaction mechanism and 564 kinetics for levoglucosan emitted in biomass burning, Chemosphere, 93, 2004-2010. http://dx.doi.org/10.1016/j.chemosphere.2013.07.021. 565 566 Bao, M., Zhang, Y.-L., Cao, F., Lin, Y.-C., Wang, Y., Liu, X., Zhang, W., Fan, M., Xie, F., Cary, R., Dixon, 567 J., Zhou, L., 2021. Highly time-resolved characterization of carbonaceous aerosols using a two-568 wavelength Sunset thermal-optical carbon analyzer, Atmospheric Measurement Techniques, 14, 4053-

4068. http://dx.doi.org/10.5194/amt-14-4053-2021.

- C. Fountoukis, &Nenes, A., 2007. ISORROPIA II: a computationally efficient thermodynamic
- equilibrium model for K<sup>+</sup>-Ca<sup>2+</sup>-Mg<sup>2+</sup>-NH<sub>4</sub><sup>+</sup>-Na<sup>+</sup>-SO<sub>4</sub><sup>2-</sup>-NO<sub>3</sub><sup>-</sup>-Cl<sup>-</sup>-H<sub>2</sub>O aerosols, Atmos. Chem. Phys., 571
- 7, 4639-4659. http://dx.doi.org/10.5194/acp-7-4639-2007.
- Charles, &J.Stone, 1985. Additive regression and other nonparametric models, The Annals of Statistics,
- 13, 689-705.
- Chen, G., Canonaco, F., Tobler, A., Aas, W., Alastuey, A., Allan, J., Atabakhsh, S., Aurela, M., Baltensperger,
- U., Bougiatioti, A., De Brito, J.F., Ceburnis, D., Chazeau, B., Chebaicheb, H., Daellenbach, K.R., Ehn, M., El
- Haddad, I., Eleftheriadis, K., Favez, O., Flentje, H., Font, A., Fossum, K., Freney, E., Gini, M., Green,
- D.C., Heikkinen, L., Herrmann, H., Kalogridis, A.C., Keernik, H., Lhotka, R., Lin, C., Lunder,
- C., Maasikmets, M., Manousakas, M.I., Marchand, N., Marin, C., Marmureanu, L., Mihalopoulos,
- 580 N., Mocnik, G., Necki, J., O'Dowd, C., Ovadnevaite, J., Peter, T., Petit, J.E., Pikridas, M., Matthew Platt,
- S., Pokorna, P., Poulain, L., Priestman, M., Riffault, V., Rinaldi, M., Rozanski, K., Schwarz, J., Sciare,
- 582 J., Simon, L., Skiba, A., Slowik, J.G., Sosedova, Y., Stavroulas, I., Styszko, K., Teinemaa, E., Timonen, 583 H., Tremper, A., Vasilescu, J., Via, M., Vodicka, P., Wiedensohler, A., Zografou, O., Cruz Minguillon,
- 584 M., Prevot, A.S.H., 2022. European aerosol phenomenology - 8: Harmonised source apportionment of
- organic aerosol using 22 Year-long ACSM/AMS datasets, Environ. Int., 166, 107325.
- http://dx.doi.org/10.1016/j.envint.2022.107325.
- Chen, J., Li, C., Ristovski, Z., Milic, A., Gu, Y., Islam, M.S., Wang, S., Hao, J., Zhang, H., He, C., Guo, H., Fu,
- H., Miljevic, B., Morawska, L., Thai, P., Lam, Y.F., Pereira, G., Ding, A., Huang, X., Dumka, U.C., 2017. A
- review of biomass burning: Emissions and impacts on air quality, health and climate in China, Sci. Total
- Environ., 579, 1000-1034. http://dx.doi.org/10.1016/j.scitotenv.2016.11.025.
- Cheng, Y., Cao, X.B., Liu, J.M., Yu, Q.Q., Zhong, Y.J., Geng, G.N., Zhang, Q., He, K.B., 2022. New open
- burning policy reshaped the aerosol characteristics of agricultural fire episodes in Northeast China, Sci.
- Total Environ., 810, 152272. http://dx.doi.org/10.1016/j.scitotenv.2021.152272.
- Cheng, Y., Engling, G., He, K.B., Duan, F.K., Ma, Y.L., Du, Z.Y., Liu, J.M., Zheng, M., Weber, R.J., 2013.
- Biomass burning contribution to Beijing aerosol, Atmos. Chem. Phys., 13, 7765-7781.
- http://dx.doi.org/10.5194/acp-13-7765-2013. 596
- D. Hoffman, A.T., Y. Iinuma, and H. Herrmann, 2010. Atmospheric stability of levoglucosan A detailed
- laboratory and modeling study, Environ. Sci. Technol., 44, 694-699. http://dx.doi.org/10.1021/es902476f.
- Donahue, N.M., Robinson, A.L., Hartz, K.E.H., Sage, A.M., Weitkamp, E.A., 2005. Competitive oxidation
- in atmospheric aerosols: The case for relative kinetics, Geophys. Res. Lett., 32.
- http://dx.doi.org/10.1029/2005gl022893.
- Engling, G., Lee, J.J., Tsai, Y.-W., Lung, S.-C.C., Chou, C.C.K., Chan, C.-Y., 2009. Size-Resolved
- Anhydrosugar Composition in Smoke Aerosol from Controlled Field Burning of Rice Straw, Aerosol Sci.
- Technol., 43, 662-672. http://dx.doi.org/10.1080/02786820902825113. 604
- Fabbri, D., Torri, C., Simoneit, B.R.T., Marynowski, L., Rushdi, A.I., Fabiańska, M.J., 2009. Levoglucosan
- and other cellulose and lignin markers in emissions from burning of Miocene lignites, Atmos. Environ.,
- 43, 2286-2295. http://dx.doi.org/10.1016/j.atmosenv.2009.01.030.
- Fu, P.Q., Kawamura, K., Chen, J., Li, J., Sun, Y.L., Liu, Y., Tachibana, E., Aggarwal, S.G., Okuzawa,
- 609 K., Tanimoto, H., Kanaya, Y., Wang, Z.F., 2012. Diurnal variations of organic molecular tracers and stable
- carbon isotopic composition in atmospheric aerosols over Mt. Tai in the North China Plain: an influence
- of biomass burning, Atmos. Chem. Phys., 12, 8359-8375. http://dx.doi.org/10.5194/acp-12-8359-2012.
- Haque, M.M., Zhang, Y., Bikkina, S., Lee, M., Kawamura, K., 2022. Regional heterogeneities in the
- emission of airborne primary sugar compounds and biogenic secondary organic aerosols in the East Asian

- outflow: evidence for coal combustion as a source of levoglucosan, Atmos. Chem. Phys., 22, 1373-1393.
- http://dx.doi.org/10.5194/acp-22-1373-2022.
- He, X., Wang, Q., Huang, X.H.H., Huang, D.D., Zhou, M., Qiao, L., Zhu, S., Ma, Y., Wang, H., Li, L., Huang,
- C., Xu, W., Worsnop, D.R., Goldstein, A.H., Yu, J.Z., 2020. Hourly measurements of organic molecular
- markers in urban Shanghai, China: Observation of enhanced formation of secondary organic aerosol
- during particulate matter episodic periods, Atmos. Environ., 240.
- <a href="http://dx.doi.org/10.1016/j.atmosenv.2020.117807">http://dx.doi.org/10.1016/j.atmosenv.2020.117807</a>.
- Hennigan, C.J., Izumi, J., Sullivan, A.P., Weber, R.J., Nenes, A., 2015. A critical evaluation of proxy
- methods used to estimate the acidity of atmospheric particles, Atmos. Chem. Phys., 15, 2775-2790.
- http://dx.doi.org/10.5194/acp-15-2775-2015.
- Hennigan, C.J., Sullivan, A.P., Collett, J.L., Robinson, A.L., 2010. Levoglucosan stability in biomass
- burning particles exposed to hydroxyl radicals, Geophys. Res. Lett., 37.
- http://dx.doi.org/10.1029/2010g1043088.
- Hong, Y., Cao, F., Fan, M.-Y., Lin, Y.-C., Gul, C., Yu, M., Wu, X., Zhai, X., Zhang, Y.-L., 2022. Impacts of
- chemical degradation of levoglucosan on quantifying biomass burning contribution to carbonaceous
- aerosols: A case study in Northeast China, Sci. Total Environ., 819
- http://dx.doi.org/10.1016/j.scitotenv.2021.152007.
- Hu, C., Wei, Z., Zhan, H., Gu, W., Liu, H., Chen, A., Jiang, B., Yue, F., Zhang, R., Fan, S., He, P., Leung,
- 632 K.M.Y., Wang, X., Xie, Z., 2022. Molecular characteristics, sources and influencing factors of isoprene
- and monoterpenes secondary organic aerosol tracers in the marine atmosphere over the Arctic Ocean,
- Sci. Total Environ., 853, 158645. http://dx.doi.org/10.1016/j.scitotenv.2022.158645.
- Huff Hartz, K.E., Weitkamp, E.A., Sage, A.M., Donahue, N.M., Robinson, A.L., 2007. Laboratory
- measurements of the oxidation kinetics of organic aerosol mixtures using a relative rate constants
- approach, J. Geophys. Res.:Atmos., 112. <a href="http://dx.doi.org/10.1029/2006jd007526">http://dx.doi.org/10.1029/2006jd007526</a>.
- Kang, M., Ren, L., Ren, H., Zhao, Y., Kawamura, K., Zhang, H., Wei, L., Sun, Y., Wang, Z., Fu, P., 2018.
- Primary biogenic and anthropogenic sources of organic aerosols in Beijing, China: Insights from
- saccharides and n-alkanes, Environ. Pollut., 243, 1579-1587.
- http://dx.doi.org/10.1016/j.envpol.2018.09.118.
- Karavoltsos, S., Sakellari, A., Bakeas, E., Bekiaris, G., Plavsic, M., Proestos, C., Zinelis, S., Koukoulakis,
- 643 K., Diakos, I., Dassenakis, M., Kalogeropoulos, N., 2020. Trace elements, polycyclic aromatic
- hydrocarbons, mineral composition, and FT-IR characterization of unrefined sea and rock salts:
- environmental interactions, Environ. Sci. Pollut. Res. Int., 27, 10857-10868.
- <u>http://dx.doi.org/10.1007/s11356-020-07670-2</u>.
- Kim, N., Park, M., Yum, S.S., Park, J.S., Song, I.H., Shin, H.J., Ahn, J.Y., Kwak, K.-H., Kim, H., Bae, G.-
- 648 N., Lee, G., 2017. Hygroscopic properties of urban aerosols and their cloud condensation nuclei activities
- measured in Seoul during the MAPS-Seoul campaign, Atmos. Environ., 153, 217-232.
- <a href="http://dx.doi.org/10.1016/j.atmosenv.2017.01.034">http://dx.doi.org/10.1016/j.atmosenv.2017.01.034</a>.
- Kumar, V., Rajput, P., Goel, A., 2018. Atmospheric abundance of HULIS during wintertime in Indo-
- Gangetic Plain: impact of biomass burning emissions, J. Atmos. Chem., 75, 385-398.
- http://dx.doi.org/10.1007/s10874-018-9381-4.
- Kuo, L.J., Louchouarn, P., Herbert, B.E., 2011. Influence of combustion conditions on yields of solvent-
- extractable anhydrosugars and lignin phenols in chars: implications for characterizations of biomass
- combustion residues, Chemosphere, 85, 797-805. http://dx.doi.org/10.1016/j.chemosphere.2011.06.074.
- Lai, C., Liu, Y., Ma, J., Ma, Q., He, H., 2014. Degradation kinetics of levoglucosan initiated by hydroxyl

- radical under different environmental conditions, Atmos. Environ., 91, 32-39.
- http://dx.doi.org/10.1016/j.atmosenv.2014.03.054.
- Li, Q., Zhang, K., Li, R., Yang, L., Yi, Y., Liu, Z., Zhang, X., Feng, J., Wang, Q., Wang, W., Huang, L., Wang,
- Y., Wang, S., Chen, H., Chan, A., Latif, M.T., Ooi, M.C.G., Manomaiphiboon, K., Yu, J., Li, L., 2023a.
- Underestimation of biomass burning contribution to PM<sub>2.5</sub> due to its chemical degradation based on
- hourly measurements of organic tracers: A case study in the Yangtze River Delta (YRD) region, China,
- Sci. Total Environ., 872. <a href="http://dx.doi.org/10.1016/j.scitotenv.2023.162071">http://dx.doi.org/10.1016/j.scitotenv.2023.162071</a>.
- Li, R., Wang, Q., He, X., Zhu, S., Zhang, K., Duan, Y., Fu, Q., Qiao, L., Wang, Y., Huang, L., Li, L., Yu, J.Z.,
- 2020. Source apportionment of PM<sub>2.5</sub> in Shanghai based on hourly organic molecular markers and other
- source tracers, Atmos. Chem. Phys., 20, 12047-12061. http://dx.doi.org/10.5194/acp-20-12047-2020.
- Li, R., Zhang, K., Li, Q., Yang, L., Wang, S., Liu, Z., Zhang, X., Chen, H., Yi, Y., Feng, J., Wang, Q., Huang,
- 669 L., Wang, W., Wang, Y., Yu, J.Z., Li, L., 2023b. Characteristics and degradation of organic aerosols from
- cooking sources based on hourly observations of organic molecular markers in urban environments,
- Atmos. Chem. Phys., 23, 3065-3081. <a href="http://dx.doi.org/10.5194/acp-23-3065-2023">http://dx.doi.org/10.5194/acp-23-3065-2023</a>.
- Li, Y., Fu, T.-M., Yu, J.Z., Feng, X., Zhang, L., Chen, J., Boreddy, S.K.R., Kawamura, K., Fu, P., Yang, X., Zhu,
- 673 L., Zeng, Z., 2021. Impacts of Chemical Degradation on the Global Budget of Atmospheric Levoglucosan
- and Its Use As a Biomass Burning Tracer, Environ. Sci. Technol., 55, 5525-5536.
- http://dx.doi.org/10.1021/acs.est.0c07313.
- Liang, L., Engling, G., Du, Z., Cheng, Y., Duan, F., Liu, X., He, K., 2016. Seasonal variations and source
- estimation of saccharides in atmospheric particulate matter in Beijing, China, Chemosphere, 150, 365-
- 377. http://dx.doi.org/10.1016/j.chemosphere.2016.02.002.
- Liu, X., Zhang, Y.-L., Peng, Y., Xu, L., Zhu, C., Cao, F., Zhai, X., Haque, M.M., Yang, C., Chang, Y., Huang,
- T.,Xu, Z.,Bao, M.,Zhang, W.,Fan, M.,Lee, X., 2019. Chemical and optical properties of carbonaceous
- aerosols in Nanjing, eastern China: regionally transported biomass burning contribution, Atmos. Chem.
- Phys., 19, 11213-11233. http://dx.doi.org/10.5194/acp-19-11213-2019.
- Mochida, M., Kawamura, K., Fu, P., Takemura, T., 2010. Seasonal variation of levoglucosan in aerosols
- over the western North Pacific and its assessment as a biomass-burning tracer, Atmos. Environ., 44, 3511-
- 3518. http://dx.doi.org/10.1016/j.atmosenv.2010.06.017.
- Pan, R., Zhu, J., Chen, D., Cheng, H., Huang, L., Wang, Y., Li, L., 2025. Integrated analysis of air quality-
- vegetation-health effects of near-future air pollution control strategies, Environ. Pollut., 366, 125407.
- <u>http://dx.doi.org/10.1016/j.envpol.2024.125407</u>.
- Pio, C.A., Legrand, M., Alves, C.A., Oliveira, T., Afonso, J., Caseiro, A., Puxbaum, H., Sanchez-Ochoa,
- 690 A., Gelencsér, A., 2008. Chemical composition of atmospheric aerosols during the 2003 summer intense
- forest fire period, Atmos. Environ., 42, 7530-7543. http://dx.doi.org/10.1016/j.atmosenv.2008.05.032.
- Pio, C.A., Legrand, M., Oliveira, T., Afonso, J., Santos, C., Caseiro, A., Fialho, P., Barata, F., Puxbaum,
- H.,Sanchez-Ochoa, A.,Kasper-Giebl, A.,Gelencsér, A.,Preunkert, S.,Schock, M., 2007. Climatology of
- aerosol composition (organic versus inorganic) at nonurban sites on a west-east transect across Europe,
- 695 J. Geophys. Res.:Atmos., 112. http://dx.doi.org/10.1029/2006jd008038.
- Sang, X., Zhang, Z., Chan, C., Engling, G., 2013. Source categories and contribution of biomass smoke to
- organic aerosol over the southeastern Tibetan Plateau, Atmos. Environ., 78, 113-123.
- <u>http://dx.doi.org/10.1016/j.atmosenv.2012.12.012.</u>
- Slade, J.H., & Knopf, D.A., 2014. Multiphase OH oxidation kinetics of organic aerosol The role of particle
- phase state and relative humidity, Geophys. Res. Lett., 41, 5297-5306.
- http://dx.doi.org/10.1002/2014GL060582.

- St John, P.C., Guan, Y., Kim, Y., Etz, B.D., Kim, S., Paton, R.S., 2020a. Quantum chemical calculations for
- over 200,000 organic radical species and 40,000 associated closed-shell molecules, Sci Data, 7, 244.
- http://dx.doi.org/10.1038/s41597-020-00588-x.
- St John, P.C., Guan, Y., Kim, Y., Kim, S., Paton, R.S., 2020b. Prediction of organic homolytic bond
- dissociation enthalpies at near chemical accuracy with sub-second computational cost, Nat. Commun.,
- 11, 2328. http://dx.doi.org/10.1038/s41467-020-16201-z.
- Stevens, H.,Barmuta, L.A.,Chase, Z.,Saunders, K.M.,Zawadzki, A.,Bowie, A.R.,Perron, M.M.G.,Sanz
- Rodriguez, E., Paull, B., Child, D.P., Hotchkis, M.A.C., Proemse, B.C., 2024. Comparing levoglucosan and
- mannosan ratios in sediments and corresponding aerosols from recent Australian fires, Sci. Total Environ.,
- 945, 174068. http://dx.doi.org/10.1016/j.scitotenv.2024.174068.
- Vicente, E.D., Vicente, A., Evtyugina, M., Carvalho, R., Tarelho, L.A.C., Oduber, F.I., Alves, C., 2018.
- Particulate and gaseous emissions from charcoal combustion in barbecue grills, Fuel Process. Technol.,
- 176, 296-306. http://dx.doi.org/10.1016/j.fuproc.2018.03.004.
- Wang, Q.,He, X.,Zhou, M.,Huang, D.D.,Qiao, L.,Zhu, S.,Ma, Y.-g.,Wang, H.-l.,Li, L.,Huang, C.,Huang,
- X.H.H., Xu, W., Worsnop, D., Goldstein, A.H., Guo, H., Yu, J.Z., 2020. Hourly Measurements of Organic
- Molecular Markers in Urban Shanghai, China: Primary Organic Aerosol Source Identification and
- Observation of Cooking Aerosol Aging, ACS Earth Space Chem., 4, 1670-1685.
- http://dx.doi.org/10.1021/acsearthspacechem.0c00205.
- Wang, Q., Wang, S., Chen, H., Zhang, Z., Yu, H., Chan, M.N., Yu, J.Z., 2025. Ambient Measurements of
- Daytime Decay Rates of Levoglucosan, Mannosan, and Galactosan, J. Geophys. Res.:Atmos., 130.
- http://dx.doi.org/10.1029/2024jd042423.
- Wang, Q., &Yu, J.Z., 2021. Ambient Measurements of Heterogeneous Ozone Oxidation Rates of Oleic,
- Elaidic, and Linoleic Acid Using a Relative Rate Constant Approach in an Urban Environment, Geophys.
- Res. Lett., 48. <a href="http://dx.doi.org/10.1029/2021gl095130">http://dx.doi.org/10.1029/2021gl095130</a>.
- White, W.H., 2008. Chemical markers for sea salt in IMPROVE aerosol data, Atmos. Environ., 42, 261-
- 274. http://dx.doi.org/10.1016/j.atmosenv.2007.09.040.
- Xu, S.,Ren, L.,Lang, Y.,Hou, S.,Ren, H.,Wei, L.,Wu, L.,Deng, J.,Hu, W.,Pan, X.,Sun, Y.,Wang, Z.,Su,
- H., Cheng, Y., Fu, P., 2020. Molecular markers of biomass burning and primary biological aerosols in
- urban Beijing: size distribution and seasonal variation, Atmos. Chem. Phys., 20, 3623-3644.
- http://dx.doi.org/10.5194/acp-20-3623-2020.
- Yan, C., Sullivan, A.P., Cheng, Y., Zheng, M., Zhang, Y., Zhu, T., Collett, J.L., 2019. Characterization of
- saccharides and associated usage in determining biogenic and biomass burning aerosols in atmospheric
- fine particulate matter in the North China Plain, Sci. Total Environ., 650, 2939-2950.
- <a href="http://dx.doi.org/10.1016/j.scitotenv.2018.09.325">http://dx.doi.org/10.1016/j.scitotenv.2018.09.325</a>.
- Yan, C., Zheng, M., Sullivan, A.P., Shen, G., Chen, Y., Wang, S., Zhao, B., Cai, S., Desyaterik, Y., Li, X., Zhou,
- 737 T., Gustafsson, Ö., Collett, J.L., 2018. Residential Coal Combustion as a Source of Levoglucosan in China,
- Environ. Sci. Technol., 52, 1665-1674. http://dx.doi.org/10.1021/acs.est.7b05858.
- Yi, Y.,Li, R.,Zhang, K.,Yang, X.,Li, Q.,Geng, C.,Chen, H.,Yang, W.,Yu, J.Z.,Li, L., 2024. Insights Into
- the Influence of Anthropogenic Emissions on the Formation of Secondary Organic Aerosols Based on
- Online Measurements, J. Geophys. Res.:Atmos., 129. <a href="http://dx.doi.org/10.1029/2024jd041479">http://dx.doi.org/10.1029/2024jd041479</a>.
- Zhai, S., Jacob, D.J., Wang, X., Shen, L., Li, K., Zhang, Y., Gui, K., Zhao, T., Liao, H., 2019. Fine particulate
- matter (PM2.5) trends in China, 2013–2018: separating contributions from anthropogenic emissions and
- meteorology, Atmos. Chem. Phys., 19, 11031-11041. http://dx.doi.org/10.5194/acp-19-11031-2019.
- Zhang, K., Yang, L., Li, Q., Li, R., Zhang, D., Xu, W., Feng, J., Wang, Q., Wang, W., Huang, L., Yaluk,

## https://doi.org/10.5194/egusphere-2025-5481 Preprint. Discussion started: 18 November 2025 © Author(s) 2025. CC BY 4.0 License.

- E.A., Wang, Y., Yu, J.Z., Li, L., 2021a. Hourly measurement of PM<sub>2.5</sub>-bound nonpolar organic compounds
- in Shanghai: Characteristics, sources and health risk assessment, Sci. Total Environ., 789.
- <u>http://dx.doi.org/10.1016/j.scitotenv.2021.148070</u>.
- Zhang, L., Zhou, P., Zhong, H., Zhao, Y., Dai, L., Wang, Q.g., Xi, M., Lu, Y., Wang, Y., 2021b. Quantifying
- the impacts of anthropogenic and natural perturbations on gaseous elemental mercury (GEM) at a
- suburban site in eastern China using generalized additive models, Atmos. Environ., 247.
- <a href="http://dx.doi.org/10.1016/j.atmosenv.2020.118181">http://dx.doi.org/10.1016/j.atmosenv.2020.118181</a>.
- Zhang, Q.,Jimenez, J.L.,Canagaratna, M.R.,Allan, J.D.,Coe, H.,Ulbrich, I.,Alfarra, M.R.,Takami,
- 754 A., Middlebrook, A.M., Sun, Y.L., Dzepina, K., Dunlea, E., Docherty, K., DeCarlo, P.F., Salcedo, D., Onasch,
- 755 T., Jayne, J.T., Miyoshi, T., Shimono, A., Hatakeyama, S., Takegawa, N., Kondo, Y., Schneider, J., Drewnick,
- F.,Borrmann, S.,Weimer, S.,Demerjian, K.,Williams, P.,Bower, K.,Bahreini, R.,Cottrell, L.,Griffin,
- R.J., Rautiainen, J., Sun, J.Y., Zhang, Y.M., Worsnop, D.R., 2007. Ubiquity and dominance of oxygenated
- species in organic aerosols in anthropogenically-influenced Northern Hemisphere midlatitudes, Geophys.
- Res. Lett., 34. http://dx.doi.org/10.1029/2007g1029979.
- Zhang, R.,Jing, J.,Tao, J.,Hsu, S.C.,Wang, G.,Cao, J.,Lee, C.S.L.,Zhu, L.,Chen, Z.,Zhao, Y.,Shen, Z.,
- 2013. Chemical characterization and source apportionment of PM<sub>2.5</sub> in Beijing: seasonal perspective,
- Atmos. Chem. Phys., 13, 7053-7074. http://dx.doi.org/10.5194/acp-13-7053-2013.
- Zhao, R., Mungall, E.L., Lee, A.K.Y., Aljawhary, D., Abbatt, J.P.D., 2014. Aqueous-phase photooxidation
- of levoglucosan a mechanistic study using aerosol time-of-flight chemical ionization mass
- spectrometry (Aerosol ToF-CIMS), Atmos. Chem. Phys., 14, 9695-9706. http://dx.doi.org/10.5194/acp-
- 14-9695-2014.
- Zhao, Y.,Kreisberg, N.M.,Worton, D.R.,Isaacman, G.,Weber, R.J.,Liu, S.,Day, D.A.,Russell,
- L.M., Markovic, M.Z., VandenBoer, T.C., Murphy, J.G., Hering, S.V., Goldstein, A.H., 2013. Insights into
- Secondary Organic Aerosol Formation Mechanisms from Measured Gas/Particle Partitioning of Specific
- Organic Tracer Compounds, Environ. Sci. Technol., 47, 3781-3787. http://dx.doi.org/10.1021/es304587x.
- Zhu, S., Wang, Q., Qiao, L., Zhou, M., Wang, S., Lou, S., Huang, D., Wang, Q., Jing, S., Wang, H., Chen,
- C., Huang, C., Yu, J.Z., 2021. Tracer-based characterization of source variations of PM<sub>2.5</sub> and organic
- carbon in Shanghai influenced by the COVID-19 lockdown, Faraday Discuss., 226, 112-137.
- http://dx.doi.org/10.1039/d0fd00091d.
