# Peer review of "Degradation of anhydro-saccharides and the driving factors"

_EGUsphere, 2025_

## Referee Comment (RC2)

Review of egusphere-2025-5481

This paper presents results from near real-time measurements of anhydrosugars made at 3 sites in China during autumn and winter. The data is used to examine the decay rate of the anhydrosugars at each site. A generalized additive model was employed to examine the parameters influencing the decay rate.

Overall, this is a good paper. As anhydrosugars are often used as biomass burning markers, understanding their decay in the atmosphere is important. But I do feel the authors don't provide enough details on the differences between the sites, how that could play a role in the ratios observed/fuels burned/burn practices, and what separates a day with a linear decay rate to one without. Also, often what is being plotted is not accurately described in the figure caption. I have tried to note and make suggestions about these along with number of other items. These are all outlined below in the specific comments which need to be addressed before this paper can be considered for publication.

Specific Comments:
Line 64 – I am not sure what the phrase universal of this phenomenon means. Should it be universal impact of this phenomenon?

Line 85 – Suggest changing during cold season to during various cold seasons

Lines 94-95 – Suggest changing during autumn and winter season across three typical cities in the three regions to during multiple autumn and winter season deployments in three different typical cities in three regions

Line 103 – Suggest removing the comma after February

Line 117 – Suggest adding a the before Changzhou

Line 119 – Suggest adding an a before WXT520, and a before BAM1020, a the before beta-ray, and an and before NOx

Line 120 – Suggest adding an a before MODEL 49i, an and before MODEL 450i, and an a before RT-4

Line 121 – Suggest adding an an before ADI2080

Line 122 - Suggest removing the The before Solar

Line 126 – Suggest adding an a before MODEL and an and before NOx

Line 127 – Suggest adding an a before MODEL 49i, an and before MODEL 42i, a respectively before OC/EC, and an a before MODEL ECOC-610

Line 129 – Suggest adding an a before MODEL S611

Line 130 – Suggest adding a the before Hong Kong

Line 131 – Suggest adding an a before Model 5030i

Line 132 – Suggest adding an an before ADI2080

Line 134- Suggest adding an and before NOx

Lines 134-135 – The authors note in Hong Kong that the meteorological parameters, O3, and NOx were measured by AWS tower. Is that the name of the instrument or the location? It seems like that might be a location and the name of the instruments are missing.

Line 135 – Suggest adding an an before Xact

Line 136 – Suggest adding an an before X-ray

Line 139 – Suggest adding an A before Detailed and a the before TAG

Line 141 – Suggest changing observation to observations and adding an a before deuterium-labeled

Line 155 – The authors note here the species used in there ISOROPPIA runs to calculate pH and ALWC. But I don't recall measurements of NH3 being mentioned in the site description and field observation section. Aren't measurements of HNO3 also needed in the calculation?

Line 179 – The V after Kumar can be removed and a period should be added after et al as part of the citation

Line 188 – The citation should be written as Wang et al. (2025)

Lines 262-263 – The phrase while the total potassium (total K) data is relatively complete can be removed from the sentence as it has already been previously stated

Line 314 – Suggest removing the was after LST

Lines 320-322 – The authors mention that the ratio of levoglucosan to potassium can be an indicator of the aging degree of biomass burning. But levoglucosan and potassium don't have to be correlated. If there is or isn't a relationship it is often based on the type of fuel being burned and the type of burning. All the measurements were made in autumn and winter, but the sites have very different characteristics. So, couldn't there be some regional influences on this? I think the authors allude to this in the previous section and Figure S3. While the data from all sites clumps together in Figure S3, it is also on a log scale, so it covers a large range of ratios. Although the sites are described in section 2.1, the authors don't go into the regional differences other than mention of the impacts they see from the monsoon. It would be helpful to provide

these additional details.  Maybe it would also be helpful to the reader to present time series of levoglucosan, potassium, and temperature along with the diurnal profile as the diurnal pattern could be driven by "special" days.

Lines 324-326 – Suggest removing this sentence as it has already been previously stated

Line 327 – Detailed information on what can be found in the previously published paper?

Lines 328-329 – I am not sure what the authors mean here by calculated formula and the range of (2)-(7) being noted.

Line 331 – Suggest changing galactosan and K+ to galactosan with K+

Line 333 - Suggest changing galactosan and K+ to galactosan with K+

Line 334 – I am not sure what the authors mean here by calculation formulas

Figure 2
-In caption suggest adding the phrase (left column) after galactosan and (right column) after ratios
-It is surprising in plot a in the left column that galactosan is higher than mannosan.  Is this correct?  Do the authors know why this is for Zibo?

Line 339 – Suggest adding a the before three

Lines 362-364 – The authors note that not all days showed good linear fitting and this could be due to the direct emissions and transmission of biomass burning.  Are there any other characteristics that could be different across the sites and various study periods?  More warmer days observed?  Burning practices the same at each location?  Fuel type at each location?  It would be helpful to provide more context for the reader.

Lines 365-367 – The authors note how many days at each site could be fit with a linear decay pattern.  But they don't really mention anything else about the days without this pattern.  What is different about the days that don't have a linear decay?

Line 389 – Suggest adding to the end of the sentence the phrase which we explore in the next section

Line 391 – Suggest change sampling points to sampling sites

Line 429 – I believe the reference is missing

Figure 5
-In caption suggest adding a the before three
-In caption the parameters O3 through T are denoted by letters, but they are actually all shown together on plot b

Line 459 – Suggest changing Fig. 6(a) show to Fig. 6(a) shows

Line 476 – mannosan and galactosan are misspelled

Figure 6
-In caption the labels noted do not match what are in the plots, suggest changing text from degradation rate of to (a) levoglucosan, (b) mannosan, and (c) galactosan analyzed as a function of ALWC, T, RH, O3, and SSR using the GAM model.

Lines 481-488 and Figure 7 – In this section the authors note and show that ALWC, RH, and Ox show a positive correlation with levoglucosan degradation rate from the factors tested. But aren't T, SSR, and Ox naturally linked to begin with? Ozone formation is related to temperature and sunlight. In addition, RH and ALWC are not the exact same thing, but are quite similar. I guess I am not totally following how there can't already be a relationship among these factors whether or not levoglucosan decays?

Line 488 – Should promoting be prominent?

Line 524 – Suggest changing substance to species

Lines 597- 598 – I believe the format of this reference is not the same as the others as it lists the authors first initial then last name

Supplemental Information
Line 51 – Suggest changing use the stock to used the stock and changing standard to standards. A period is also missing from the end of the sentence.

Line 53 – There should be commas before and after respectively. The chemical abbreviations used are not defined. Suggest adding a the before final working

Line 54 – Suggest changing are showed to that are shown

Line 56 – Suggest changing with range to ranging

Lines 56-57 – Suggest removing the phrase (ranging from 5 to 25 uL)

Table S2
-In the caption should the word identified be added after compounds?
-For the solvent and quantification IS are they the same for all 3 anhydrosugars?

Tables S4
-In caption suggest adding the phrase average +/- standard deviation of before hourly concentrations

Table S6

-In the caption the authors call these the smoothing function parameters, but I don't believe they have been called that elsewhere in the text. Maybe it would be helpful to note what equation they refer to?
-Should it be SSR or SSRD? I believe throughout the text it was SSR.

Figure S3
-In caption the way it is written does not match what is actually plotted. Suggest changing correlation between K+bb and decay rates of levoglucosan, mannosan, and galactosan to correlation of levoglucosan, mannosan, and galactosan as a function of K+bb

Figure S4
-I believe the y-axes labels are not correct. It looks like the decay rate of levoglucosan is missing and the decay rate of galactosan is used twice.

Figure S5
-In caption the way part of it is written does not match what is actually plotted. Suggest changing predicted values vs. residuals to residuals vs. predicted values and predicted values vs. observed values to observed values vs. predicted values. Also suggest changing for GAM to for the GAM model